# Lethal Arrhythmogenic Role of Left Ventricular Myocardial Interstitial Fibrosis in Apolipoprotein E/Low-Density Lipoprotein Receptor Double-Knockout Mice with Metabolic Dysfunction-Associated Steatohepatitis

**DOI:** 10.3390/ijms26010144

**Published:** 2024-12-27

**Authors:** Jinyao Liu, Yumiko Oba, Yosuke Kondo, Ryo Nakaki, Seiko Yamano

**Affiliations:** 1Student Medical Academia Investigation Lab, Yamaguchi University Graduate School of Medicine, Ube 755-8505, Japan; 2Advanced Medical Research Academic-Course, Yamaguchi University School of Medicine, Ube 755-8505, Japan; i019eb@yamaguchi-u.ac.jp; 3Rhelixa, Inc., Tokyo 104-0042, Japan; yosuke.kondo@rhelixa.com (Y.K.); nakaki@rhelixa.com (R.N.); 4Life Science Division, Yamaguchi University Advanced Technology Institute, Ube 755-8505, Japan; yamano@yamaguchi-u.ac.jp

**Keywords:** lethal arrhythmia, metabolic dysfunction-associated steatotic liver disease, alcohol, low-carbohydrate, high-protein, high-fat atherogenic diet, left ventricular myocardial interstitial fibrosis

## Abstract

The combination of alcohol and a low-carbohydrate, high-protein, high-fat atherogenic diet (AD) increases the risk of lethal arrhythmias in apolipoprotein E/low-density lipoprotein receptor double-knockout (AL) mice with metabolic dysfunction-associated steatotic liver disease (MASLD). This study investigates whether left ventricular (LV) myocardial interstitial fibrosis (MIF), formed during the progression of metabolic dysfunction-associated steatohepatitis (MASH), contributes to this increased risk. Male AL mice were fed an AD with or without ethanol for 16 weeks, while age-matched AL and wild-type mice served as controls. Liver and heart tissues were analyzed, and susceptibility to lethal arrhythmias was assessed through histopathology, fluorescence immunohistochemistry, RNA-Seq, RT-PCR, and lethal arrhythmia-evoked test. Ethanol combined with an AD significantly induced LV MIF in MASH-affected AL mice, as shown by increased fibrosis-related gene expression, Sirius-Red staining, and elevated *collagen 1a1* and *3a1* mRNA levels, alongside a higher incidence of lethal arrhythmias. Cardiac myofibroblasts exhibited sympathetic activation and produced elevated levels of fibrosis-promoting factors. This study highlights the role of cardiac myofibroblasts in LV MIF, contributing to an increased incidence of lethal arrhythmias in MASH-affected AL mice fed ethanol and AD, even after the alcohol was fully metabolized on the day of consumption.

## 1. Introduction

Cardiovascular diseases are responsible for up to 17 million annual deaths worldwide, with approximately 25% attributed to sudden cardiac death [1]. Additionally, metabolic dysfunction-associated steatotic liver disease (MASLD) affects around 25% of the global adult population [2,3]. Both sudden cardiac death and MASLD pose distinct yet interconnected health challenges, with implications for global mortality and public health [3,4,5]. While sudden cardiac death primarily relates to sudden, unforeseen death due to lethal arrhythmias; MASLD, formerly known as non-alcoholic fatty liver disease, manifests as increased hepatocellular fat deposition, often occurring alongside metabolic dysfunctions such as obesity, insulin resistance, and dyslipidemia. MASLD without significant liver damage, defined by hepatocellular fat deposition exceeding 5%, is relatively benign and non-progressive. However, metabolic dysfunction-associated steatohepatitis (MASH), characterized by liver inflammation with bridging fibrosis, is associated with elevated mortality rates, affecting both hepatic (end-stage liver disease and hepatocellular carcinoma) and extra-hepatic (particularly cardiovascular) outcomes [6,7,8].

Previous studies have explored the common risk factors shared between sudden cardiac death and MASLD, highlighting the intricate relationship between cardiovascular and hepatic pathophysiology. Lifestyle factors like unhealthy dietary habits, alcohol intake, and obesity are identified as significant contributors to the onset and progression of both conditions [3,4,5,9]. Research endeavors have aimed to elucidate the complex mechanisms linking cardiovascular disease, metabolic dysfunction, and MASLD, with obesity, insulin resistance, and dyslipidemia identified as common pathways leading to adverse outcomes in both cardiovascular and hepatic health. Furthermore, emerging evidence suggests that MASLD serves as an independent risk factor for cardiovascular events and mortality [10].

Since 2018, there has been increasing attention to the relationship between alcohol consumption, health risks, and cardiovascular disease. According to the interpretation by GBD 2016 Alcohol Collaborators, it has been suggested that minimizing health loss requires zero alcohol consumption [11]. Chronic ethanol intake, even if fully metabolized on the day of consumption, can worsen MASLD when combined with a low-carbohydrate, high-protein, high-fat atherogenic diet (AD) [12]. Additionally, it can accelerate the development of atherosclerosis [13], promote cardiac sympathetic predominance, and increase susceptibility to lethal arrhythmias in apolipoprotein E/low-density lipoprotein receptor double-knockout (AL) mice [14]. However, the complex pathophysiology underlying the risk of sudden cardiac death during the initiation and progression of MASLD remains incompletely understood.

This passage discusses the relationship between gap junctional remodeling, interstitial fibrosis, and impaired electrical conduction velocity, which can lead to lethal arrhythmias in nonischemic heart failure [15]. It also highlights the arrhythmogenic role of the up-regulated left ventricular (LV) myocardial gap junction protein alpha 1 (Gja1, also known as Cx43) observed during the lethal arrhythmia-evoked test in AL mice with MASLD [14]. This study aims to investigate the impact of LV myocardial interstitial fibrosis (MIF) on lethal arrhythmogenicity in AL mice with MASH.

## 2. Results

### 2.1. Characteristics of the Mice

Male mice were divided into four groups as AL mice fed an AD with and without ethanol (AL+AD, Et (+) and AL+AD, Et (−)). Age-matched male AL and wild-type (WT) mice were fed a standard chow diet without ethanol (AL+SCD, Et (−) and WT+SCD, Et (−)). The sample sizes were as follows: AL+AD, Et (+): 7; AL+AD, Et (−): 8; AL+SCD, Et (−): 7; WT+SCD, Et (−): 8.

Body weights increased in both groups of mice fed the AD, regardless of ethanol supplementation (Figure 1A). After 16 weeks of consuming the co-diet of AD with ethanol, significant increases in liver (Figure 1B) and LV weights (Figure 1C) were observed in the AL mice.

The lethal arrhythmia test caused sudden cardiac death in two AL mice fed an AD-ethanol co-diet, making liver sampling impossible due to the inability to perform systemic bleeding. However, these mice were still included in the body weight measurements (Figure 1A) and LV weight measurements (Figure 1C), resulting in the difference in sample sizes (*n* = 5 for Figure 1B and *n* = 7 for Figure 1A,C) within the AL+AD, Et (+) group. Unlike liver sampling, LV sampling is less affected by residual blood resulting from the inability to perform systemic bleeding, as heart tissue is less influenced by blood pooling compared to liver tissue. Therefore, while liver sampling was not feasible, LV weight analysis was successfully performed in these mice.

Alcohol consumption was 116.2 ± 22.4 g/week/kg BW (Appendix A), and the blood ethanol concentration was 0.075 ± 0.09 mg/mL (Appendix A) at the end of the lethal arrhythmia-evoked test following 16 weeks of consuming the co-diet of AD and ethanol. This indicates that ethanol was nearly completely metabolized on the same day it was ingested.

### 2.2. MASH Induced by Addition of Ethanol to AD

Both AD with and without ethanol induced hepatic steatosis (Figure 2A,B). However, only the co-administration of AD with ethanol led to liver inflammation and bridging fibrosis. This was characterized by an elevated level of the leukocyte subset marker CD68 in immunostained liver sections (Figure 2A,C) and an increased expression of *Cd68* mRNA observed in RT-PCR (Figure 2D), reflected by the immunostained liver sections (Figure 2A,C) as “CD68+ area (%)”. Moreover, bridging fibrosis was evident in Sirius-Red-stained liver sections, with fibrosis extending across lobules between portal areas or between portal areas and central veins (Figure 2A). Additionally, there was an augmentation in Sirius-Red content (Figure 2E) and an upregulation of *collagen (Col) 1a1* mRNA expression (Figure 2F), reflected by the histological staining (Figure 2C) as “Sirius-Red content (%)”.

The results showed that *Cd68* mRNA expression was higher than the CD68-positive area detected by immunostaining in AL+SCD, Et (−) and AL+AD, Et (−). Similarly, *Col 1a1* mRNA expression was high in AL+AD, Et (−), while Sirius-Red staining did not show corresponding results. The discrepancy between mRNA expression and protein levels or histological staining can be attributed to various factors. For instance, PCR amplifies and detects even minimal levels of *Cd68* mRNA with high sensitivity, whereas the CD68-positive area identified by immunostaining depends on the availability of the protein, antibody binding efficiency, and the generation of sufficient fluorescence signals. These can be influenced by low protein abundance or structural changes in the tissue. Additionally, *Col 1a1* mRNA expression may reflect an early transcriptional response, whereas collagen deposition, as detected by Sirius-Red staining, is a later event. Therefore, Sirius-Red staining might not yet capture the ongoing transcriptional activity.

### 2.3. Increased Susceptibility to Lethal Arrhythmia in AL Mice with MASH

The lethal arrhythmia-evoked test elicited the highest incidences of lethal arrhythmias in AL mice subjected to a co-diet of AD and ethanol-induced MASH (Appendix A: 13% in WT+SCD, Et (−), 14% in AL+SCD, Et (−), 13% in AL+AD, Et (−), and 100% in AL+AD, Et (+); χ2 = 18.27, *p* = 0.0004 by contingency table chi-square test). Furthermore, the lethal arrhythmia-evoked test led to two cases of sudden cardiac death in AL mice fed the co-diet of AD with ethanol (Figure 3).

### 2.4. Ethanol and AD Additively Upregulated LV Gene Expression Related to Fibrosis in Mice with a Co-Diet of Ethanol and AD-Induced MASH

To understand how a co-diet of ethanol with AD contributed to the increased incidences of lethal arrhythmias during the lethal arrhythmia-evoked test in AL mice with MASH, we investigated gene expression profiles in the LV by RNA-Seq analysis. The 1153 genes changed by the co-diet of ethanol with AD were enriched in the functions of the collagen trimer, collagen-containing extracellular matrix, collagen type I trimer, extracellular space, and extracellular matrix by GO enrichment analysis (Figure 4A), all of which are implicated in the fibrosis associated with LV MIF pathogenesis.

Genes related to fibrosis in LV, changed by the co-diet of ethanol and AD, including myocardial *Col 1a1* and *Col 3a1*, which are associated with lethal arrhythmia onset, were determined by the RNA-Seq analysis and are shown in heatmaps (Figure 4B).

### 2.5. LV MIF in Mice with a Co-Diet of Ethanol and AD-Induced MASH

Significant LV MIF was evident, characterized by fibrosis surrounding individual cardiomyocytes or groups of cardiomyocytes in Sirius-Red-stained LV sections (Figure 5A). This was accompanied by an increase in Sirius-Red content (Figure 5B) and an upregulation of mRNA expression of LV fibrosis markers *Col 1a1* (Figure 5C) and *Col 3a1* (Figure 5D) observed in RT-PCR. In AL mice with a co-diet of ethanol and AD-induced MASH, there was a significant 6.3-fold increase in Sirius-Red content, along with 5.4-fold and 2.3-fold increases in *Col 1a1* and *Col 3a1* mRNA expressions, respectively (all *p* < 0.05 compared to those of the WT+SCD, Et (−) mice).

### 2.6. Co-Diet of Ethanol and AD Induced Features of LV Sympathetic Activation in Cardiac Myofibroblasts, Which in Turn Led to Upregulated Productions of Cardiac Myofibroblast-Derived TGF-β1

AL mice that received a combination diet of ethanol and AD for 16 weeks displayed elevated sympathetic activation in the LV compared to the other three groups. This was evidenced by an increase in the sympathetic activation marker (TH) observed in immunostained LV sections (Figure 6A,B) along with an upregulation of the sympathetic activation marker (*Npy*) mRNA expression seen in RT-PCR (Figure 7A). Specifically, there was a 7.5-fold increase in TH-positive areas, coupled with a 3.2-fold upregulation in *Npy* mRNA expression, both statistically significant compared to those of the WT+SCD, EtOH (−) mice. Additionally, the combination diet of ethanol and AD resulted in increased co-localization of TH (a sympathetic activation marker, purple, emission maximum = 670 nm) and α-SMA (an activated cardiac myofibroblast marker, green, emission maximum = 519 nm), appearing as pink, as well as increased co-localization of α-SMA and TGF-β1 (associated with fibrotic diseases, red, emission maximum = 570 nm), appearing as yellow. This is demonstrated by the heightened TH-, α-SMA-, and TGF-β1-positive areas observed in the same visual fields of immunostained LV sections (Figure 6A). This observation was supported by the upregulation of *Npy* (Figure 7A), *Acta2* (also known as α-SMA, Figure 7B), and *Tgfb1* (Figure 7C) mRNA expressions in RT-PCR. Specifically, there were 7.5-fold, 7.5-fold, and 6.5-fold increases in TH-, α-SMA-, and TGF-β1-positive areas, respectively, along with 3.2-fold, 1.9-fold, and 1.5-fold upregulation of *Npy*, *Acta2*, and *Tgfb1* mRNA expressions, respectively; all were statistically significant compared to the values for the WT+SCD, Et (−) mice.

## 3. Discussion

This study represents the first evidence of the lethal arrhythmogenic role of LV MIF in apolipoprotein E/low-density lipoprotein receptor double-knockout mice with MASH. Another significant finding is the observation of LV sympathetic activation alongside LV MIF in AL mice fed a co-diet of ethanol and AD for 16 weeks. Moreover, LV cardiac myofibroblasts, exhibiting sympathetic activation, displayed increased production of fibrosis factors. Taken together, these results suggest that LV cardiac myofibroblasts with sympathetic activation may contribute to the onset and progression of LV MIF, potentially leading to heightened susceptibility to lethal arrhythmias in AL mice with a co-diet of AD and ethanol-induced MASH, even when the alcohol was fully metabolized on the day of consumption.

Sudden cardiac death is a significant concern in public health, representing a substantial portion of cardiovascular-related mortality with high lifetime risk. Identifying and addressing modifiable risk factors are essential strategies to tackle this issue [4,16]. MASLD, encompassing a spectrum from steatosis to steatohepatitis and cirrhosis, is increasingly prevalent [17,18] and associated with cardiac complications, including sudden cardiac death [19,20,21]. Clinical guidelines recommend cardiovascular screening for MASLD patients [22,23] due to its links with atherosclerosis, LV dysfunction, and susceptibility to lethal arrhythmias, particularly in animal models with a co-diet of ethanol and atherogenic diets [13,14]. However, direct assessments of structural abnormalities like LV MIF in mouse models are limited.

LV MIF, characterized by excessive fibrous tissue within the myocardial interstitium, is implicated in impaired electrical conduction velocity and lethal arrhythmias [24,25,26]. Moreover, chronic toxicity, such as anthracyclines chemotherapy [27], and metabolic disturbances, such as diabetes and obesity [28], can also induce fibrotic changes in the myocardium. This study underscores how chronic ethanol and atherogenic diet exposure induce LV MIF, evidenced by increased Sirius-Red content in LV sections and upregulated expression of fibrosis-related genes. Moreover, LV MIF correlates with heightened susceptibility to lethal arrhythmias, including sudden cardiac death, in mice with MASH. These findings shed light on the intricate relationship between metabolic liver dysfunction and cardiac pathology, providing insights into potential mechanisms underlying the increased sudden cardiac death risk in MASLD.

The process of LV MIF often initiates with cardiomyocyte death, but various stimuli, including metabolic injury, can instigate fibrosis even without cell death. Multiple cell types, including myofibroblasts, are implicated in the fibrotic response, either by directly producing fibrous tissue or indirectly through the secretion of profibrotic mediators [29]. Factors such as physical and chemical stimuli, along with fibrogenic growth factors like TGF-β1 found in the injured myocardium [29], can trigger the generation of myofibroblasts. These myofibroblasts are pivotal in both reparative and reactive fibrosis. Upon activation, fibroblasts proliferate and transform into a non-proliferative secretory phenotype known as myofibroblasts. These cells exhibit ultrastructural and phenotypic characteristics resembling smooth muscle cells, including the formation of contractile stress fibers and the expression of α-smooth muscle actin [30,31]. In our study, we observed increased co-localization of TH (sympathetic activation marker) and α-SMA (activated cardiac myofibroblast marker) and co-localization of α-SMA and TGF-β1 (associated with fibrotic diseases) in LV sections of mice exposed to a co-diet of ethanol and AD. This suggests that cardiac myofibroblasts, activated by sympathetic stimulation, contribute to the initiation and progression of LV MIF in mice with MASH. Myofibroblasts play a crucial role in collagen fibrillogenesis and scar tissue formation during tissue repair, regardless of the underlying cause of injury or tissue type. Their secretome includes pro-collagen types I and III, essential molecules for regulating extracellular fibrillary collagen turnover. Even after the completion of healing, myofibroblasts can persist, leading to ongoing collagen turnover and progressive structural remodeling of the affected organ, such as fibrosis or sclerosis [32].

Our current study findings are consistent with our prior research [12,14], indicating that the combination of ethanol and AD increases the susceptibility of AL mice to both lethal arrhythmias and MASH. Ethanol supplementation exacerbates cardiac sympathetic activation when combined with the AD, leading to heightened activity of myofibroblasts and excessive production of fibrosis factors, thereby initiating and exacerbating LV MIF. These new findings, in conjunction with our prior discovery that ethanol supplementation to AD elevates the expression of the gap junction channel *Gja1* mRNA [14], contributing to an augmented risk of lethal arrhythmias during the lethal arrhythmia-evoked test, underscore the crucial roles played by MASH-related LV MIF and the upregulation of *Gja1* mRNA expression in the development of lethal arrhythmias. Modulating both *Gja1* mRNA expression and the LV MIF state may offer a novel avenue for mitigating lethal arrhythmogenicity induced by the combination of ethanol and AD. Additionally, our results suggest that alcohol exacerbates the progression of MASLD and increases susceptibility to lethal arrhythmias in AL mice fed AD, even when ethanol is completely metabolized on the same day of ingestion.

Our findings, supported by Frangogiannis’s review [33], suggest that MASH and cardiac fibrosis/arrhythmogenesis are linked through multiple mechanisms, notably sympathetic activation in cardiac myofibroblasts. Chronic hepatic inflammation in MASH triggers systemic inflammatory responses and sympathetic activation, driving cardiac fibroblast differentiation into myofibroblasts. These cells remodel the extracellular matrix, leading to fibrosis, which disrupts electrical conduction and heightens arrhythmia risk. Additionally, catecholamines from sympathetic activation may exacerbate arrhythmogenesis by altering ion channel function. Exploring these pathways could uncover therapeutic targets. Furthermore, the findings linking MASH to cardiac fibrosis and arrhythmogenesis highlight the clinical relevance of TGF-β as a key mediator in fibrotic diseases, with implications for potential therapeutic targets [33,34]. TGF-β drives fibrosis in both the heart and liver by activating fibroblasts and promoting their differentiation into myofibroblasts, which deposit collagen and contribute to fibrosis. In MASH, TGF-β exacerbates liver injury and fibrosis, impacting both structure and function [33,34]. Chronic liver inflammation in MASH activates the sympathetic nervous system, which in turn enhances TGF-β signaling, promoting fibrosis. This interaction between sympathetic activation and TGF-β signaling is crucial for the development of both fibrosis and arrhythmias, suggesting that targeting both pathways could benefit treatment [33]. Inhibiting TGF-β signaling using drugs like pirfenidone and losartan may help reduce fibrosis in both the liver and heart [34], and sympathetic blockers such as β-blockers (e.g., carvedilol) can mitigate fibrosis and arrhythmias by reducing sympathetic activation [33]. However, clear evidence of a causative link between liver and heart pathologic events is lacking. Exploring the causal relationship between MASLD and cardiovascular diseases, including lethal arrhythmias, and the link between hepatic inflammation and cardiac fibrosis focusing on cardiac myofibroblasts in LV MIF, warrant further exploration. Collecting liver and heart tissues at defined time points (e.g., 4, 8, and 12 weeks) for histological and molecular analysis to track inflammatory and fibrotic markers such as TGF-β1, α-SMA, and collagen deposition could be used to elucidate the temporal relationship between hepatic inflammation and cardiac fibrosis in a mouse model with MASH and LV MIF.

## 4. Materials and Methods

The research was conducted following the ethical principles outlined in the U.S. National Institutes of Health Guide for the Care and Use of Laboratory Animals (NIH Publication Eighth Edition) and received approval from the Institutional Animal Care and Use Committee of Yamaguchi University (permit no. 77-001).

### 4.1. Mouse Model and Diets

The experimental mice used in the study were sourced as frozen embryos of AL mice, specifically the B6.129-Apoetm1Unc Ldlrtm1Her/J strain (#002246), obtained from Jackson Laboratory. These embryos were then revived by Charles River Japan in Osaka, Japan, and subsequently bred at the Institutional Animal Care and Use Committee of Yamaguchi University. It is important to note that only male mice were included in the experimental groups.

Male mice aged twelve weeks were divided into four experimental groups as follows: AL mice fed an AD fine-powder diet (containing 12% sucrose [carbohydrate], 52% casein [protein], and 21% butter [fat]; product No.: 7685-130119 from Oriental Bio Service, Kyoto, Japan) with and without ethanol (designated as AL+AD, Et (+) and AL+AD, Et (−), respectively). Age-matched male AL and C57BL/j (Charles River Japan, Osaka, Japan) wild-type (WT) mice fed a standard chow diet (SCD: a commercial pelleted diet comprising 55.3% carbohydrate, 23.1% protein, and 5.1% fat; Oriental Yeast Co., Ltd., Tokyo, Japan) without ethanol (designated as AL+SCD, Et (−) and WT+SCD, Et (−), respectively).

To acclimatize the mice to ethanol treatment, they were initially provided with a 5 g/dL ethanol-water solution for the first week, followed by a 10 g/dL ethanol-water solution for the subsequent 15 weeks, totaling 16 weeks of treatment. The AD fine-powder diet was administered using a powder feeder, while ethanol (JIS special grade, purity greater than 99.5%, Sigma-Aldrich, Inc., Tokyo, Japan) was offered separately from AD, mixed with water in a drinking bottle. Both AD and ethanol were available ad libitum. Weekly monitoring was conducted to track changes in the mice’s body weight (BW) and their consumption of diets (SCD or AD) and drink (water or ethanol).

### 4.2. Lethal Arrhythmia-Evoked Test

Following the conclusion of the study, an electrocardiographic (ECG) transmitter from Transoma Medical Data Sciences International, St. Paul, MN, USA, was implanted beneath the abdominal skin of the mice. This was accompanied by subcutaneous electrodes arranged in a lead II configuration. The procedure was conducted under anesthesia with 1.0–1.5% (*v*/*v*) isoflurane in oxygen. Subsequently, an intraperitoneal infusion of epinephrine (2 mg/kg BW) was administered after the 4 h acute restraint stress (ARS) period, also defined as the lethal arrhythmia-evoked test in the present study, as detailed in prior research [14]. These procedures were conducted on unanesthetized mice, and continuous ECG telemetry recordings were obtained from 30 mice (WT+SCD, Et (−): *n* = 8; AL+SCD, Et (−): *n* = 7; AL+AD, Et (−): *n* = 8; AL+AD, Et (+): *n* = 7). Lethal arrhythmias, including complete atrioventricular block (CAVB), ventricular tachycardia (VT), ventricular fibrillation (VF), and asystole, were defined for analysis. CAVB, also known as third-degree atrioventricular block, indicates a complete failure of supraventricular impulses to reach the ventricles, resulting in independent atrial and ventricular activities. VT was characterized by the presence of more than 6 consecutive ventricular extrasystoles, while VF was identified by the loss of ECG synchronicity and decreased amplitude, as described by Liu [14]. Asystole is when the heart is completely still, and there is no waveform on ECG.

### 4.3. Blood, Liver, and LV Tissues Collection

Following the lethal arrhythmia-evoked test, blood samples were taken from the LV. Liver and LV tissues were collected after systemic perfusion with phosphate-buffered saline (PBS) under 1.0–1.5% (*v*/*v*) isoflurane anesthesia in oxygen. Some tissue samples were immersed in RNA*later* (a Qiagen RNA stabilization reagent, Tokyo, Japan) and stored at −80 °C. The remaining tissues were fixed in a 4% paraformaldehyde (PFA) solution in 0.1 M PBS, washed with 15% sucrose, and stored at 4 °C. It is important to note that in the AL+AD, Et (+) group, two mice died during the lethal arrhythmia-evoked test, resulting in the inability to collect liver samples from these individuals.

### 4.4. Histopathological and Fluorescence Immunohistochemical Examination

Liver and LV samples underwent fixation using 4% PFA in 0.1 M PBS followed by embedding in an optimal cutting temperature compound and slicing into 5-μm-thick frozen sections for subsequent histopathological and fluorescence immunohistochemical analyses.

Histological assessments of liver and LV tissues were conducted using Oil-Red-O and Sirius-Red staining techniques to examine changes. Stained samples were observed under a microscope (BZ-X800; Keyence, Osaka, Japan) to quantify the stained areas. Analysis was performed on samples from 5 visual fields at 20× magnification to evaluate fat deposition and fibrosis.

Immunofluorescence staining utilized specific primary antibodies, including CD68 (SCB sc-9139, rabbit polyclonal antibody, 1:200) for liver, alpha smooth muscle actin (α-SMA; #E2464, rabbit polyclonal antibody, 1:200; Spring Bioscience, Pleasanton, CA, USA), transforming growth factor-β1 (TGF-β1; SCB sc-31608, goat polyclonal antibody, 1:200), and tyrosine hydroxylase (TH; #NBP3-05555, chicken polyclonal antibody, 1:200; Novus Biologicals, Centennial, CO, USA) for LV. Subsequently, the samples underwent secondary antibody incubation. Alexa Fluor^®^ 488 AffiniPure alpaca anti-rabbit IgG (#611-545-215, 1:500; Jackson ImmunoResearch Laboratories, West Grove, PA, USA) was used for CD68 and α-SMA, Cy^TM3^-conjugated AffiniPure donkey anti-goat IgG for TGF-β1 (#705-165-147, 1:500), and Cy^TM5^-conjugated AffiniPure donkey anti-chicken for TH (#703-175-155). Finally, the samples were mounted in a VECTASHIELD PLUS antifade mounting medium with DAPI (#H-1200; Vector Laboratories, Newark, NJ, USA) for nuclear staining. Stained samples were examined under a confocal microscope (STELLARIS 8; Leica Microsystems GmbH, Wetzlar, Germany), and positive areas for CD68, α-SMA, TGF-β1, and TH were quantified using ImageJ software (2.3.0/1.53o). This comprehensive analysis aimed to assess inflammation in liver tissue and evaluate cardiac myofibroblasts, TGF-β1 expression, and sympathetic activation in LV tissue.

### 4.5. Real-Time Reverse Transcriptase Polymerase Chain Reaction (RT-PCR)

Total RNA extraction was carried out on both liver and LV tissues using the RNeasy^®^ Mini Kit and RNeasy^®^ Fibrous Tissue Mini Kit (Qiagen, Tokyo, Japan). This process included the elimination of genomic DNA and subsequent synthesis of complementary DNA (cDNA) following the manufacturer’s protocols (ReverTra Ace^®^ qPCR RT Master Mix with gDNA Remover; Toyobo, Osaka, Japan). RT-PCR was performed using the Applied Biosystems StepOnePlus^™^ system (Applied Biosystems, Foster City, CA, USA). Specific assays were utilized for mRNA expression analysis, targeting genes such as *Cd68* (Assay ID: Mm03047340_m1), *collagen 1a1* (*Col 1a1*; Assay ID: Mm00801666_g1), *collagen 3a1* (*Col 3a1*; Assay ID: Mm01254476_m1), *actin alpha 2* (*Acta2*; Assay ID: Mm01546133_m1; also known as α-SMA), *transforming growth factor beta 1* (*Tgfb1*; Assay ID: Mm01178820_m1), *neuropeptide Y* (*Npy*; Assay ID: Mm00445771_m1), and *glyceraldehyde-3-phosphate dehydrogenase* (*Gapdh*; Assay ID: Mm99999915_g1). Normalization of these mRNA expressions was performed against *Gapdh* mRNA levels within the same cDNA sample, utilizing a comparative quantitative approach. The results are presented as fold changes relative to the WT+SCD, Et (−) group.

### 4.6. RNA-Seq and Pathway Analyses

LV total RNA samples were sent to Rhelixa Inc. (Tokyo, Japan) and sequenced with Illumina NovaSeq 6000 (Illumina, Inc., San Diego, CA, USA) with a paired-end sequencing length of 150 bp for approximately 4G bases per sample by Strand-specific RNA-Seq with NEBNext^®^ Poly (A) mRNA Magnetic Isolation Module (E7490, New England Biolabs Inc., Ipswich, MA, USA) and NEBNext^®^ UltraTMII Directional RNA Library Prep Kit (E7760, New England Biolabs Inc.).

The quality of the raw paired-end sequence reads was assessed with FastQC (Version 0.11.7; https://www.bioinformatics.babraham.ac.uk/projects/fastqc/ (accessed on 12 July 2023). Low quality (<20) bases and adapter sequences were trimmed by Trimmomatic software (Version 0.38) with the following parameters: ILLUMINACLIP: path/to/adapter.fa:2:30:10, LEADING:20, TRAILING:20, SLIDINGWINDOW:4:15, and MINLEN:36. Then, the trimmed reads were aligned to the reference mouse genome (Version mm 10) using RNA-Seq aligner HISAT2 (Version 2.1.0). Finally, the raw read counts were normalized with transcripts per million (TPM).

The raw read counts were normalized by relative log normalization (RLE), and differentially expressed analysis was conducted with DESeq2 (Version 1.24.0). Gene ontology (GO) enrichment analysis was performed with GOATOOLS (Version 1.1.6). To assess the results of the RNA-Seq analysis in terms of cellular components, the genes that changed in the mice LV myocardium were matched to the genes related to each GO team, and the 10 terms with lowest *p*-value were listed.

### 4.7. Analysis of Blood Ethanol Level at the End of the Lethal Arrhythmia-Evoked Test

The plasma ethanol level at the conclusion of the lethal arrhythmia-evoked test was determined using a specific saliva alcohol testing kit (QED A150; product code #31150, ToxTests, Dayton, OH, USA) following the protocol outlined in the study by Shiraishi et al. [7].

### 4.8. Statistical Analyses

Continuous data were presented as mean ± standard deviation (SD). Outliers were identified and removed using the Smirnov–Grubbs test. The occurrence of lethal arrhythmia was evaluated using a contingency table (m × n) chi-square test. If the *p* value from the chi-square test was significant, multiple comparisons among the four groups were conducted using the Bonferroni–Dunn post hoc test. One-way ANOVA was employed to compare continuous variables among the four groups, followed by the Bonferroni–Dunn post hoc test if the *F* value was significant. Statistical analyses were performed using Statcel2 for Windows software (Statcel2, oms-publishing, Saitama, Japan). Statistical significance was determined at *p* < 0.05.

## 5. Conclusions

In summary, this study investigated the influence of combining ethanol with an atherogenic diet (AD) on the heightened susceptibility to lethal arrhythmias in a mouse model with metabolic dysfunction-associated steatohepatitis (MASH). The results revealed that the presence of both ethanol and AD led to an increase in left ventricular myocardial interstitial fibrosis (LV MIF), which was associated with a higher incidence of lethal arrhythmias, including sudden cardiac death during the lethal arrhythmia-evoked test. These findings emphasize the importance of avoiding simultaneous consumption of alcohol with nutritionally imbalanced diets, as it can significantly impact the risk of lethal arrhythmias, even when the alcohol was fully metabolized on the day of consumption. This highlights the necessity for public health interventions aimed at preventing sudden cardiac death and will help establish treatment strategies for human patients with MASH and its extra-hepatic complications.

## Figures and Tables

**Figure 1 ijms-26-00144-f001:**
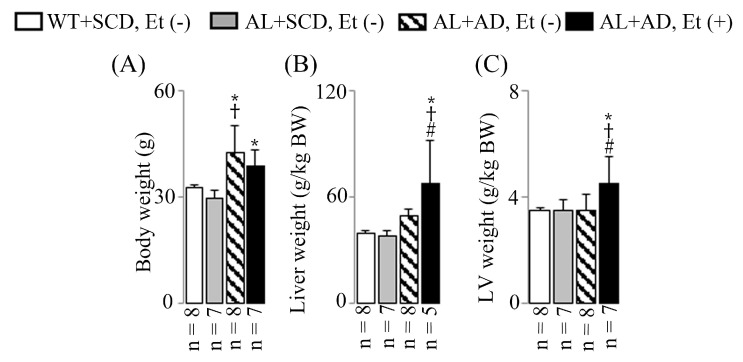
The key characteristics of the animals. Representative results for body weight (BW) (**A**), liver weight (**B**), and LV weight (**C**). The data are presented as mean values with standard deviations (SDs). Statistical significance (* *p* < 0.05 vs. WT+SCD, Et (−); ^†^ *p* < 0.05 vs. AL+SCD, EtOH (−); ^#^ *p* < 0.05 vs. AL+AD, EtOH (−)) was determined using ANOVA followed by Bonferroni–Dunn post hoc test.

**Figure 2 ijms-26-00144-f002:**
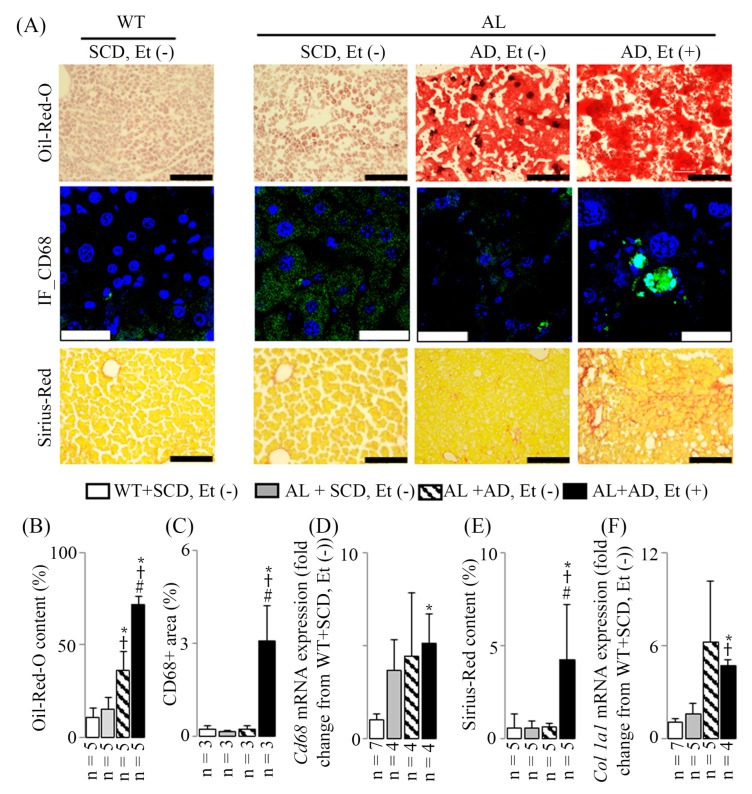
The effects of a co-diet of ethanol and AD on hepatic fat accumulation, inflammation, and fibrosis. Examples of Oil-Red-O staining, CD68 immunostaining, and Sirius-Red staining of liver sections (**A**). The scale bars are 200 μm for Oil-Red-O and Sirius-Red staining and 30 μm for CD68 immunostaining. Representative results of hepatic Oil-Red-O content (**B**), CD68-positive area (**C**), *Cd68* mRNA expression (**D**), Sirius-Red content (**E**), and *Col 1a1* mRNA expression (**F**). The data are presented as mean values with standard deviations (SDs). Statistical significance (* *p* < 0.05 vs. WT+SCD, Et (−); ^†^ *p* < 0.05 vs. AL+SCD, EtOH (−); ^#^ *p* < 0.05 vs. AL+AD, EtOH (−)) was determined using ANOVA followed by Bonferroni–Dunn post hoc test.

**Figure 3 ijms-26-00144-f003:**
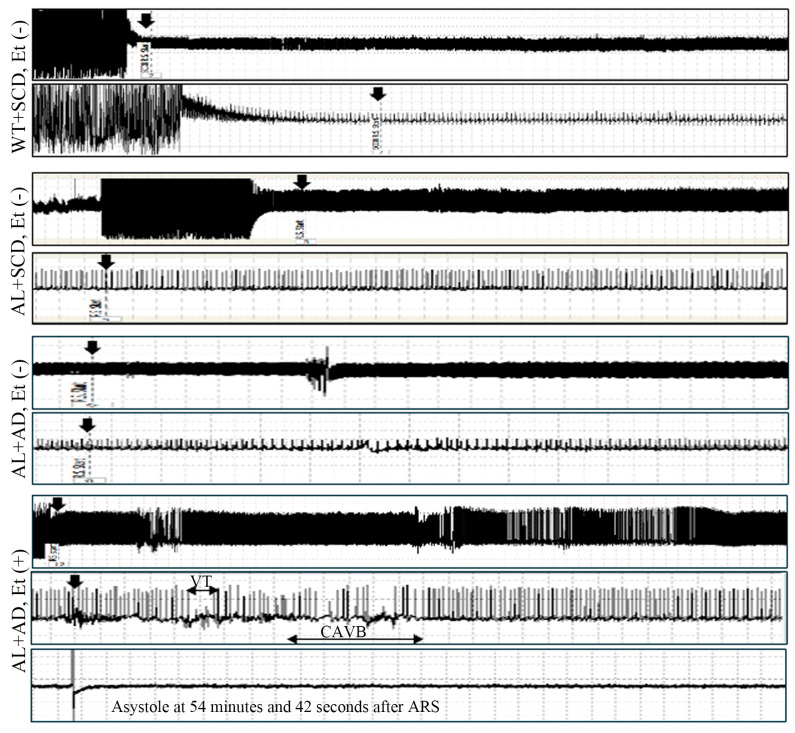
Sudden cardiac death resulting from the lethal arrhythmia-evoked test. Shown are representative electrocardiograms (ECGs) depicting the occurrence of complete atrioventricular block (CAVB), ventricular tachycardia (VT), and asystole induced by acute restraint stress (ARS) in AL mice fed with a co-diet of ethanol and AD. Each panel displays a 2 min ECG and 15 s zoomed-in lethal arrhythmia-evoked test, with an arrow indicating the onset of ARS.

**Figure 4 ijms-26-00144-f004:**
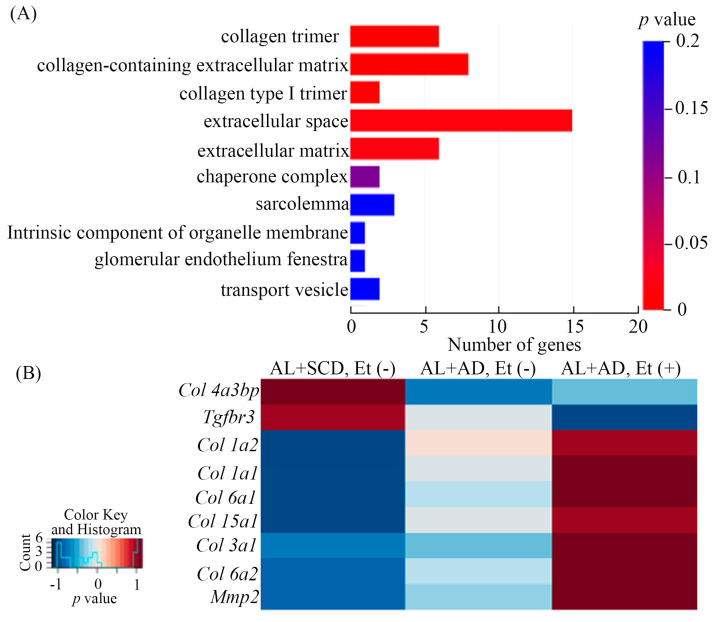
Ethanol and AD changed gene expression related to fibrosis in LV. Cellular components of the genes significantly changed in the gene ontology (GO) enrichment analysis: to assess the results of the RNA-Seq analysis in terms of cellular components, the genes that changed in mice LV myocardium were matched to the genes related to each GO term, and the 10 terms with the lowest *p*-value were listed (**A**). The heatmap shows the gene expression changes related to fibrosis in the LV, ranked by the lowest *p*-values as determined by RNA-Seq analysis (**B**).

**Figure 5 ijms-26-00144-f005:**
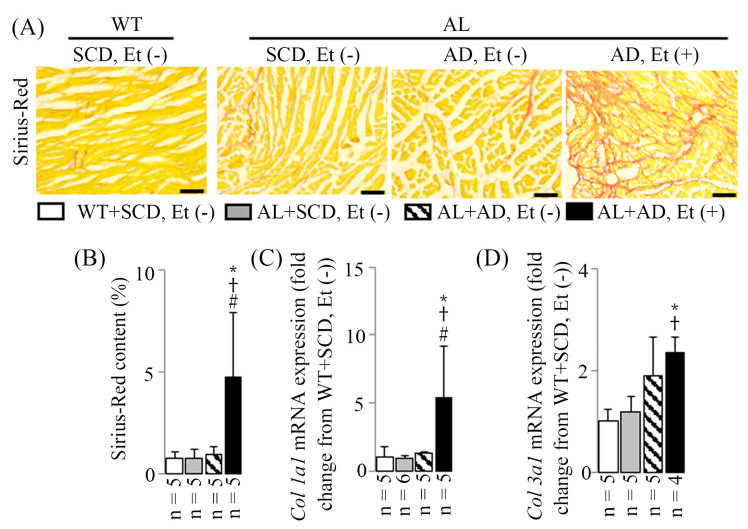
LV MIF resulting from the co-diet of ethanol and AD. Examples of LV sections stained with Sirius-Red (**A**), with a scale bar of 100 μm. Representative data of LV Sirius-Red content (**B**), LV *Col 1a1* mRNA expression (**C**), and LV *Col 3a1* mRNA expression (**D**). The data are presented as mean values with standard deviations (SDs). Statistical significance (* *p* < 0.05 vs. WT+SCD, Et (−); ^†^ *p* < 0.05 vs. AL+SCD, EtOH (−); ^#^ *p* < 0.05 vs. AL+AD, EtOH (−)) was determined using ANOVA followed by Bonferroni–Dunn post hoc test.

**Figure 6 ijms-26-00144-f006:**
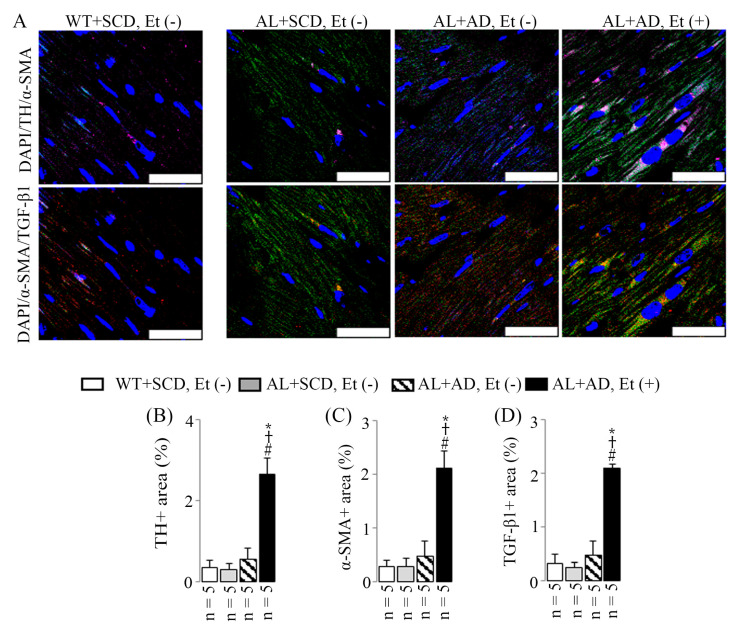
The combination diet of ethanol and AD increased the co-localization of TH (sympathetic activation marker) and α-SMA (activated cardiac myofibroblast marker) and the co-localization of α-SMA and TGF-β1 (associated with fibrotic diseases) in the immunostained LV sections. Examples merging nuclear (DAPI, blue), TH, and α-SMA appear as pink and merging nuclear (DAPI, blue), α-SMA, and TGF-β1 appear as yellow (**A**) alongside measurements of the LV-TH-positive area (**B**), α-SMA-positive area (**C**), and TGF-β1-positive area (**D**) within the same visual field of immunostained LV sections. The scale bar denotes 30 μm. Results are presented as mean ± SD. Statistical significance was determined as * *p* < 0.05 compared to WT+SCD, Et (−), ^†^ *p* < 0.05 compared to AL+SCD, EtOH (−), and ^#^ *p* < 0.05 compared to AL+AD, EtOH (−) using ANOVA followed by Bonferroni–Dunn post hoc test.

**Figure 7 ijms-26-00144-f007:**
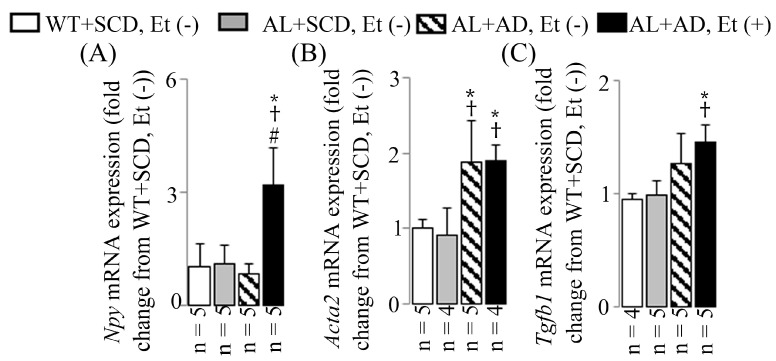
The impact of a co-diet of ethanol and AD on the mRNA expressions in the LV as determined by RT-PCR. The mRNA levels of *Npy* (**A**), *Acta2* (**B**), and *Tgfb1* (**C**) in mice subjected to the lethal arrhythmia-evoked test. The data are presented as mean values with standard deviations (SDs). Statistical significance (* *p* < 0.05 vs. WT+SCD, Et (−); ^†^ *p* < 0.05 vs. AL+SCD, EtOH (−); ^#^ *p* < 0.05 vs. AL+AD, EtOH (−)) was determined using ANOVA followed by Bonferroni–Dunn post hoc test.

## Data Availability

The datasets analyzed during this study are available from the authors upon reasonable request.

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
