# Peer review of "Lethal Arrhythmogenic Role of Left Ventricular Myocardial Interstitial Fibrosis in Apolipoprotein E/Low-Density Lipoprotein Receptor Double-Knockout Mice with Metabolic Dysfunction-Associated Steatohepatitis"

_ijms, 2024, doi:10.3390/ijms26010144_

Round 1
Reviewer 1 Report
Comments and Suggestions for Authors
The Authors demonstrate in this manuscript that an atherogenic diet combined with ethanol in apolipoprotein E/low-density lipoprotein receptor double-knockout (AL) mice is associated with the outcome of MASLD and myocardial interstitial fibrosis (MIF).
Major Revision:
- In several parts of the manuscript, the Authors seems to suggest that MIF depends on the progression of metabolic dysfunction-associated steatohepatitis (MASH). However, clear evidence of a causative link between these two pathologic events is lacking and they appear more like independent events occurring in parallel during an atherogenic diet combined with ethanol in AL mice. If the Authors wish to propose a causative link, they need to provide specific experiments or at least clear evidence from the literature.
- Variations in the Cd86 mRNA expression (Fig2D) and the Col1a1 mRNA expression (Fig2F) are not reflected by the IF and the histological staining showed in Fig2A. How the Authors explain this point?
- The figures of the CD68 staining in Fig2A are the same reported in the previous paper of the group, published in 2022, in Fig4A.
Minor comments:
- the supplementary tables and figures are not cited in the manuscript
- in Fig1B, for the AL+AD, Et+ group the n is 5, while in Fig1A and C the n is 7. The Authors excluded some values? In case, explain why.
- In Fig6A, Authors must specify the fluorescence color of the different stained proteins to make the figure more understandable.
Author Response
Responses for Comments and Suggestions for Authors_1
The Authors demonstrate in this manuscript that an atherogenic diet combined with ethanol in apolipoprotein E/low-density lipoprotein receptor double-knockout (AL) mice is associated with the outcome of MASLD and myocardial interstitial fibrosis (MIF).
Major Revision:
- In several parts of the manuscript, the Authors seems to suggest that MIF depends on the progression of metabolic dysfunction-associated steatohepatitis (MASH). However, clear evidence of a causative link between these two pathologic events is lacking and they appear more like independent events occurring in parallel during an atherogenic diet combined with ethanol in AL mice. If the Authors wish to propose a causative link, they need to provide specific experiments or at least clear evidence from the literature.
I agree with your comments and suggestions. We have added "However, clear evidence of a causative link between these two pathologic events is lacking" to lines 294 and 295.
- Variations in the Cd86 mRNA expression (Fig2D) and the Col1a1 mRNA expression (Fig2F) are not reflected by the IF and the histological staining showed in Fig2A. How the Authors explain this point?
In response to your comments, we added [reflected by the immunostained liver sections (Figure 2A, C) as “CD68+ area (%)”] to line 104-105 and [reflected by the histological staining (Fig2C) as “Sirius-Red content (%)”] to line 109-110.
- The figures of the CD68 staining in Fig2A are the same reported in the previous paper of the group, published in 2022, in Fig4A.
I apologize for not being able to confirm the mistake. I have corrected the exact photo for figures of the CD68 staining in Fig2A.
Minor comments:
- the supplementary tables and figures are not cited in the manuscript
We added “(Table S1 of Supplementary Material)” in line 94 to line 95; “(Table S2 and Figure S1 and S2 of Supplementary Material)” in line 95 to line 96; and “(Table S3 of Supplementary Material)” in line 122 to line 123. The order of the table has been rearranged also.
- in Fig1B, for the AL+AD, Et+ group the n is 5, while in Fig1A and C the n is 7. The Authors excluded some values? In case, explain why.
The lethal arrhythmia test caused sudden cardiac death in two AL mice fed an AD-ethanol co-diet, preventing liver sampling. Liver samples in this group differed by body weights (Fig. 1A) and LV weights (Fig. 1C). We added this explains in line 85-87.
- In Fig6A, Authors must specify the fluorescence color of the different stained proteins to make the figure more understandable.
We adjusted fluorescence colors for different proteins to clarify the figure.
Submission Date: 07 October 2024
Date of this review: 16 Oct 2024 18:49:37

Reviewer 2 Report
Comments and Suggestions for Authors
This manuscript presents an interesting study on the arrhythmogenic role of left ventricular myocardial interstitial fibrosis in apolipoprotein E/low-density lipoprotein receptor double-knockout mice with metabolic dysfunction-associated steatohepatitis (MASH). The authors investigate the effects of combining alcohol with a low-carbohydrate, high-protein, high-fat atherogenic diet on the risk of lethal arrhythmias. The study's strengths include:
Well-designed experimental groups and comprehensive analysis techniques, including histopathology, fluorescence immunohistochemistry, RNA-Seq, RT-PCR, and arrhythmia test.
Demonstration of MASH induction with increased susceptibility to lethal arrhythmias in AL mice fed with ethanol and AD.
Thorough examination of LV MIF through various methods, including Sirius-Red staining and gene expression analysis.
However, there are some limitations and areas for improvment:
The sample size for each group is not clearly stated, which affects the statistical power of the results.
The mechanistic link between hepatic inflammation and cardiac fibrosis could be further explored, particularly regarding the role of cardiac myofibroblasts in LV MIF.
The clinical implications for human patients with MASH are not adequately discussed.
Recommendations:
Include a more detailed discusion on the potential mechanisms linking MASH to cardiac fibrosis and arrhythmogenesis, especially considering the observed sympathetic activation in cardiac myofibroblasts.
Provide a clearer explanation of the clinical relevance and potential therapeutic targets based on these findings, particularly in relation to the role of TGF-β in fibrotic diseases.
Consider performing additional experiments to elucidate the temporal relationship between hepatic inflammation and cardiac fibrosis development.
In conclusion, while this study provides insights into the arrhythmogenic role of LV MIF in MASH, some revisions are necessary to strengthen its impact and clinical relevance.
Author Response
Responses for Comments and Suggestions for Authors_2
This manuscript presents an interesting study on the arrhythmogenic role of left ventricular myocardial interstitial fibrosis in apolipoprotein E/low-density lipoprotein receptor double-knockout mice with metabolic dysfunction-associated steatohepatitis (MASH). The authors investigate the effects of combining alcohol with a low-carbohydrate, high-protein, high-fat atherogenic diet on the risk of lethal arrhythmias. The study's strengths include:
Well-designed experimental groups and comprehensive analysis techniques, including histopathology, fluorescence immunohistochemistry, RNA-Seq, RT-PCR, and arrhythmia test.
Demonstration of MASH induction with increased susceptibility to lethal arrhythmias in AL mice fed with ethanol and AD.
Thorough examination of LV MIF through various methods, including Sirius-Red staining and gene expression analysis.
However, there are some limitations and areas for improvment:
The sample size for each group is not clearly stated, which affects the statistical power of the results.
We added “The sample sizes are AL+AD, Et (+): 7; AL+AD, Et (−): 8; AL+SCD, Et (−): 7; WT+SCD, Et (−): 8.” in line 80-81
The mechanistic link between hepatic inflammation and cardiac fibrosis could be further explored, particularly regarding the role of cardiac myofibroblasts in LV MIF.
We added a future outlook in lines 297-298: “The link between hepatic inflammation and cardiac fibrosis, focusing on cardiac myofibroblasts in LV MIF, warrants further exploration.”
The clinical implications for human patients with MASH are not adequately discussed.
We added “This will help establish treatment strategies for human patients with MASH and its extra-hepatic complications.” in line 447-448
Recommendations:
Include a more detailed discusion on the potential mechanisms linking MASH to cardiac fibrosis and arrhythmogenesis, especially considering the observed sympathetic activation in cardiac myofibroblasts.
In response to your comments, we added “Our findings, supported by Frangogiannis’s review (33), suggest that MASH and cardiac fibrosis/arrhythmogenesis are linked through multiple mechanisms, notably sympathetic activation in cardiac myofibroblasts. Chronic hepatic inflammation in MASH triggers systemic inflammatory responses and sympathetic activation, driving cardiac fibroblast differentiation into myofibroblasts. These cells remodel the extracellular matrix, leading to fibrosis, which disrupts electrical conduction and heightens arrhythmia risk. Additionally, catecholamines from sympathetic activation may exacerbate arrhythmogenesis by altering ion channel function. Exploring these pathways could uncover therapeutic target.” in line 274-282.
Provide a clearer explanation of the clinical relevance and potential therapeutic targets based on these findings, particularly in relation to the role of TGF-β in fibrotic diseases.
In response to your comments, we added “Furthermore, the findings linking MASH to cardiac fibrosis and arrhythmogenesis highlight the clinical relevance of TGF-β as a key mediator in fibrotic diseases, with implications for potential therapeutic targets. TGF-β drives fibrosis in both the heart and liver by activating fibroblasts and promoting their differentiation into myofibroblasts, which deposit collagen and contribute to fibrosis. In MASH, TGF-β exacerbates liver injury and fibrosis, impacting both structure and function (33,34). Chronic liver inflammation in MASH activates the sympathetic nervous system, which in turn enhances TGF-β signaling, promoting fibrosis. This interaction between sympathetic activation and TGF-β signaling is crucial in the development of both fibrosis and arrhythmias, suggesting that targeting both pathways could benefit treatment (33). Inhibiting TGF-β signaling, using drugs like pirfenidone and losartan, may help reduce fibrosis in both the liver and heart (34) and Sympathetic blockers such as β-blockers (e.g., carvedilol) can mitigate fibrosis and arrhythmias by reducing sympathetic activation (33).” in line 282-294
Consider performing additional experiments to elucidate the temporal relationship between hepatic inflammation and cardiac fibrosis development.
In response to your comments, we added “Collect liver and heart tissues at defined time points (e.g., 4, 8, and 12 weeks) for histological and molecular analysis to track inflammatory and fibrotic markers such as TGF-β1, α-SMA, and collagen deposition could be used to elucidate the temporal relationship between hepatic inflammation and cardiac fibrosis in mouse model with MASH and LV MIF.” in line 298-302
In conclusion, while this study provides insights into the arrhythmogenic role of LV MIF in MASH, some revisions are necessary to strengthen its impact and clinical relevance.
Submission Date
07 October 2024
Date of this review
04 Dec 2024 16:04:49

Round 2
Reviewer 1 Report
Comments and Suggestions for Authors
The Authors replied to almost all, but a couple of requests were not considered:
- Why the cd68 mRNA expression is so high in AL+SCD, Et- and AL+AD, Et- while for this two group the IF for CD68 is so week? Why the Col1a1 mRNA expression is high in AL+AD, Et- while the Sirius-red is not relevant for this group?
The sentences added by the Authors do no answered the question.
- Reains unclear why the AL+AD, Et+ group in Fig1B is 5, while in Fig1A and C the n is 7. Authors also stated at lines 81-82 that "The sample sizes are ... AL+SCD, Et (−): 7.
Authors replied that "The lethal arrhythmia test caused sudden cardiac death in two AL mice fed an 86 AD-ethanol co-diet, preventing liver sampling" but was unclear while LV weight was assesed. This is not convincing.
- In figure 6A what is the fluorescence color for TH? for a-SMA? for TGFb1? they need to be specified in the image.
Author Response
- Why the cd68 mRNA expression is so high in AL+SCD, Et- and AL+AD, Et- while for this two group the IF for CD68 is so week? Why the Col1a1 mRNA expression is high in AL+AD, Et- while the Sirius-red is not relevant for this group?
The sentences added by the Authors do no answered the question.
The discrepancy between mRNA expression and protein levels or histological staining can be explained by post-transcriptional and post-translational regulatory mechanisms, as well as differences in detection methods. We added the following explanation:
"The results showed that Cd68 mRNA expression was higher than the CD68-positive area detected by immunostaining in AL+SCD, Et (-) and AL+AD, Et (-). Similarly, Col 1a1 mRNA expression was high in AL+AD, Et (-), while Sirius-red staining did not show corresponding results. The discrepancy between mRNA expression and protein levels or histological staining can be attributed to various factors. For instance, PCR amplifies and detects even minimal levels of Cd68 mRNA with high sensitivity, whereas the CD68-positive area identified by immunostaining depends on the availability of the protein, antibody binding efficiency, and the generation of sufficient fluorescence signals. These can be influenced by low protein abundance or structural changes in the tissue. Additionally, Col 1a1 mRNA expression may reflect an early transcriptional response, whereas collagen deposition, as detected by Sirius-red staining, is a later event. Therefore, Sirius-red staining might not yet capture the ongoing transcriptional activity."
This was included in lines 118–129 to provide a comprehensive explanation of the observed discrepancies.
- Reains unclear why the AL+AD, Et+ group in Fig1B is 5, while in Fig1A and C the n is 7. Authors also stated at lines 81-82 that "The sample sizes are ... AL+SCD, Et (−): 7.
Authors replied that "The lethal arrhythmia test caused sudden cardiac death in two AL mice fed an 86 AD-ethanol co-diet, preventing liver sampling" but was unclear while LV weight was assesed. This is not convincing.
The lethal arrhythmia test caused sudden cardiac death in two AL mice fed an AD-ethanol co-diet, making liver sampling impossible due to the inability to perform systemic bleeding. However, these mice were still included in the body weight measurements (Fig. 1A) and LV weight measurements (Fig. 1C), resulting in the difference in sample sizes (n = 5 for Fig. 1B and n = 7 for Fig. 1A and Fig. 1C) within the AL+AD, Et (+) group. Unlike liver sampling, LV sampling is less affected by residual blood resulting from the inability to perform systemic bleeding, as heart tissue is less influenced by blood pooling compared to liver tissue. Therefore, while liver sampling was not feasible, LV weight analysis was successfully performed in these mice.
We have clarified this explanation in lines 86–94 to ensure consistency and transparency.
- In figure 6A what is the fluorescence color for TH? for a-SMA? for TGFb1? they need to be specified in the image.
In response to your comments, we edited the “Additionally, the combination diet of ethanol and AD resulted in increased co-localization of TH (sympathetic activation marker) and α-SMA (activated cardiac myofibroblast marker) and co-localization of α-SMA and TGF-β1 (associated with fibrotic diseases), as indicated by heightened TH-, α-SMA-, and TGF-β1-positive areas in the same visual fields of immunostained LV sections (Figure 6A).” to
“Additionally, the combination diet of ethanol and AD resulted in increased co-localization of TH (a sympathetic activation marker, purple, emission maximum = 670 nm) and α-SMA (an activated cardiac myofibroblast marker, green, emission maximum = 519 nm), appearing as pink, as well as increased co-localization of α-SMA and TGF-β1 (associated with fibrotic diseases, red, emission maximum = 570 nm), appearing as yellow. This is demonstrated by the heightened TH-, α-SMA-, and TGF-β1-positive areas observed in the same visual fields of immunostained LV sections (Figure 6A).” in lines 197–203.

Reviewer 2 Report
Comments and Suggestions for Authors
The authors are to be commended for this interesting work. My questions and comments have been addressed satisfactorily.
Author Response
The authors are to be commended for this interesting work. My questions and comments have been addressed satisfactorily.
Thank you for your valuable guidance and understanding.
